# Study on the Socio-Economic Impact of Cancer Disease on Cancer Patients and Their Relatives

**DOI:** 10.3390/healthcare10122370

**Published:** 2022-11-25

**Authors:** Alberto Garcia Martin, Eduardo J. Fernandez Rodriguez, Celia Sanchez Gomez, Maria I. Rihuete Galve

**Affiliations:** 1Department of Labour law and Social Work, University of Salamanca, 37007 Salamanca, Spain; 2Department of Nursing and Physiotherapy, University of Salamanca, 37007 Salamanca, Spain; 3Instituto de Investigación Biomédica de Salamanca (IBSAL), 37007 Salamanca, Spain; 4Department of Developmental and Educational Psychology, University of Salamanca, 37007 Salamanca, Spain; 5Medical Oncology Unit, University Hospital of Salamanca, 37007 Salamanca, Spain

**Keywords:** access to care, cancer, inequality, healthcare utilization, disparity, oncology

## Abstract

Background: Cancer is one of the most relevant social and health problems in the world. The disease entails additional costs for cancer patients and their families that are not covered by the public part of our welfare state, and which they assume themselves simply because they are ill. The main objective of this study is to identify and analyse the additional cost and socioeconomic impact of cancer disease on patients diagnosed with cancer disease and their families. Methods: Descriptive cross-sectional randomised observational epidemiological study without replacement with prevalence of cancer disease in the study base, carried out in the Medical Oncology Service of the Complejo Asistencial Universitario de Salamanca (CAUSA), Spain. Results: The study variable has been the additional cost of the cancer disease for cancer patients and their families that is not covered by our autonomous health system. Conclusions: Cancer disease entails an additional cost for the patient and family; more specifically, for 55% of the patients in the study sample, the diagnosis of cancer represents extra expenditure of between 8.38–9.67% of their annual income. Furthermore, the disability and dependence of patients does not represent an additional cost due to their levels of functionality, but it can have repercussions on the future cost of the evolution of the disease, in addition to the fact of having cancer.

## 1. Introduction

Cancer disease is one of the most relevant socio-health problems in the world. Due to its high mortality and increasing incidence [1,2,3], it represents the first cause of death in men and the second in women [4,5]. Cancer affects the medical aspects of the patient, but also generates a series of psychological, social, occupational, family and emotional conflicts in the lives of patients and their families [6,7,8,9,10]. The patient, during the oncological process, faces physical limitations and problems of autonomy caused by the disease, adaptation to new family dynamics and care roles in the patient, deteriorated labour relations and work incapacity, reduction of income at home, need for adaptations in the home, acquisition of orthopaedic material and complex emotional situations, among others.

The Spanish Association Against Cancer (AECC) states that the patient’s own work situation at the time of diagnosis is an important generator of crisis in the patient and family. This is a major conditioning factor for the consequences that the disease will have on the different aspects of their lives; the diagnosis itself can be a cause or aggravating factor in the socio-economic risk for the subsistence of the person and their family [11,12]. The disease entails additional costs for cancer patients and their families that are not covered by the public part of our welfare state, and which they bear themselves simply because they are ill. Public benefits are insufficient, and in the best of cases, they correspond to co-payments of a percentage of the additional expenses incurred by the patient and the family. The additional costs they bear are related to the purchase of medicines in pharmacies or parapharmacies, the purchase of orthopaedic equipment, home help or a patient accompaniment service, and additional costs arising from travelling to hospital for treatment.

Cancer is a very complex disease to deal with, but even more so when there are certain additional expenses that the patient and their family have to assume, due to the fact of being ill in a situation that also generates incapacity for work, sick leave and a reduction in income. The socio-economic impact suffered by the patient is known as the additional expenses borne by the patient and the family added to a situation of reduced household income as a result of having cancer [13,14,15].

Modern oncology, given the seriousness of the problem, has created the term "financial toxicity" to refer to the difficulties that the patient is encountering in his or her fight against cancer; not for medical reasons, but for economic reasons [13,14,15]. Because of all this, we believe that there is a need for interest and study of these costs and the socio-economic impact they produce.

On this basis, we hypothesized that cancer disease produces physical and psychosocial needs in many patients from the time of diagnosis and throughout the process of their illness that are not financed by the public health system and that can have an economic and emotional impact on the family nucleus.

Main objective of the study: identify and analyses the additional cost and socioeconomic impact of cancer disease on patients diagnosed with cancer disease and their families.

## 2. Materials and Methods

### 2.1. Design and Procedures

Randomised descriptive cross-sectional observational epidemiological study without replacement with prevalence of cancer disease at baseline.

### 2.2. Participants

The study participants were persons with an anatomopathological diagnosis of cancer admitted to the Medical Oncology Service of the Complejo Asistencial Universitario de Salamanca (CAUSA), Spain, receiving outpatient treatment in the Day Hospital and Radiation Oncology Service after voluntarily signing the informed consent form and fulfilling the selection criteria.

Inclusion criteria: Anatomopathological diagnosis of cancer, being an oncology patient at the Complejo Asistencial Universitario de Salamanca, Spain (CAUSA), being over 18 years of age and agreeing to participate voluntarily in this study.

Exclusion criteria: Failure to sign the informed consent document, despite having agreed to participate voluntarily in the study, having already been assessed for this study in a previous hospital admission or outpatient care and/or having a cognitive state that does not allow understanding of the study; Mini Mental score of less than 24 synonymous with mild cognitive impairment.

Withdrawal criteria: Express request for withdrawal by the patient’s family, even if they had completed the informed consent document, failure to correctly complete any of the assessment instruments required for this study.

### 2.3. Sample Size

The sample size was estimated considering the determination of the strength or power of the study, data on the size of the total population of people diagnosed with cancer disease in the province of Salamanca, Spain. The statistical parameters of confidence, the probability of occurrence or non-occurrence of the event studied with a maximum accepted estimation error.

The data considered for the calculation of the sample were:–The total population of people diagnosed with cancer disease in the last five years in Spain is 787,476, according to data from the Observatory of the Spanish Association Against Cancer in 2019 [12,15].–The total population of new diagnoses with cancer disease in Spain is 275,562, according to data from the Observatory of the Spanish Association Against Cancer of 2019 [12,15].–The total population of people diagnosed with cancer disease in the last five years in Castilla y León is 49,725, according to data from the 2019 Observatory of the Spanish Association Against Cancer [12,15].–The total population of new diagnoses with cancer disease in Castilla y León is 17,592, according to data from the Observatory of the Spanish Association Against Cancer in 2019 [12,15].–The total number of people diagnosed with cancer disease in the province of Salamanca is 7043, according to data from the Observatory of the Spanish Association Against Cancer in 2019 [12,15].–The total number of new diagnoses of cancer disease in the province of Salamanca is 2494, according to data from the Observatory of the Spanish Association Against Cancer of 2019 [12,15].

The sample size (n = …) is obtained from the result of the formula for the calculation of the sample in health research, described for descriptive studies of a qualitative type for finite population:(1)$$n=\frac{N\times z_{\alpha^{2}}\times p\times q}{e^{2}\times \left(N-1\right)+z_{\alpha^{2}}\times p\times q}$$

The different data applied in the formula determine that the optimal result of our study corresponds to a sample size of 365 patients:

Sample size n = 364.33

### 2.4. Variables

#### 2.4.1. Study Variable

The study variable was the additional cost of cancer for cancer patients and their families that is not covered by our regional health system.

#### 2.4.2. Intervening Variables

The intervening variables were: gender, age, diagnosis, stage of the disease, line of treatment of the disease, levels of dependency, comorbidities, accessibility to treatment/change of habitual residence, socioeconomic situation of the patient and/or family, employment, pensions/social security benefits, main carer overload, health-related quality of life in cancer patients (HRQoL).

### 2.5. Measuring Instruments

To assess the different variables in the study, we used the following instruments to evaluate and collect results:

Barthel Index (BI) [16]: used to measure the levels of dependency of patients. It takes into account mobility parameters in addition to the care activities of the person being assessed. It is divided into 10 items corresponding to activities of daily living ABDVD. We use this measurement scale due to the objectivity of the results and the simplicity in the collection of each of its items. Score from 0 to 100.

Lawton and Brody scale [17]: used to measure people in their adaptation to their environment and maintaining independence with their community. It takes into account parameters of ability to perform tasks involving tools and social activities and measures ability to perform instrumental activities of daily living (IADLs). It is divided into eight items. We use this scale as a complement to the IB to be able to see the adaptation in the context of the oncological patient and the capacity or not of the patient to maintain a certain degree of autonomy in his or her community environment. Scoring from 0 to 8.

ZARIT primary caregiver overload test [18]: used to assess family overload with respect to caring for the ill person. It has 22 items. It has a Cronbach’s alpha coefficient of 0.91 and test-retest reliability with a Pearson correlation coefficient of 0.86 between the two means. We use this scale for reliability and validity. Score from 0 to 88.

ECOG scale [19,20]: used to measure the quality of life of cancer patients. It takes into account the evolution of the patient’s abilities in daily life and the relationship with the patient’s own maintenance of autonomy. Validity and reliability is high; Kendall’s correlation coefficient 0.75 with high correlation between this scale and the Kamofsky Index. The Spearman correlation coefficient values obtained were 0.85 (*p* < 0.0001) and 0.87 (*p* < 000.1) in different studies. We used this scale and not another one because of the objectivity of considering the quality of life of the cancer patient. Score from 0 to 5.

EUROQOL-5D questionnaire [21,22]: used to assess the variation in health-related quality of life. It takes into account five dimensions of health: mobility, self-care, activities of daily living, pain/discomfort and anxiety/depression. We use this scale for its simplicity, time of administration and ability to measure physical, psychological and social dimensions without loss of number of responses or wrong answers.

Self-completion questionnaire created for the study: used with the aim of obtaining relevant information on the additional cost that the oncological disease produces in patients and family members, considering: patient identification data, identification data of the patient’s main caregiver, patient health data, associated comorbidities, employment situation and economic situation of the patient and family. The dimensions in turn have different items and sub-items that have been previously assessed and evaluated objectively to obtain more information.

To evaluate the intervening variable “Socioeconomic situation of patient and/or family”, and the variable under study “Additional cost of the cancer disease to the cancer patient and family”, we used the self-completion questionnaire created for the study.

To assess the intervening variables “Gender”, “Age”, “Diagnosis”, “Stage of the disease”, “Line of treatment of the disease”, “Employment; self-employed, employed”, “Pensions/Social Security benefits”, “Comorbidity” and “Accessibility to treatment/change of home” we used the self-completion questionnaire created for the study.

To assess the intervening variables “Level of dependency”, “Primary caregiver overload” and “Health-related quality of life in cancer patients (HRQoL)” we used the rest of the measurement scales: Barthel Index, Lawton-Brody Scale, Zarit overload test, ECOG Scale and EuroQol-5D Questionnaire.

### 2.6. Procedure and Data Collection

The data collection did not have a temporal sequence. It was carried out at a single point in time, in a period of time that allowed the data to be obtained. The study was conducted in accordance with the Declaration of Helsinki and approved by the Ethics Committee of University of Salamanca (protocol code 507 and date of approval February 2021).

The patients and main caregivers were informed of the existence of the present research, as well as of the objective and voluntary nature of their participation in it. They signed the informed consent form authorising their voluntary participation in the study.

The evaluation and data collection were carried out during the following phases:

New admission of the patient to the Medical Oncology Department: we analysed the suitability of the newly admitted patient according to the inclusion and exclusion requirements, presented the research and main objective, gave informed consent for authorisation to participate in the study and completed the self-completion questionnaire. Subsequently, we passed the scales.

Patient admitted when we started the study in the Medical Oncology Department: we analysed the suitability of the admitted patient, presented the research and main objective, gave informed consent for authorisation to participate in the study and completed the self-completion questionnaire. Subsequently, we passed the scales.

Treatment of patients in day hospital and/or treatment of patients in radiotherapy: we analysed the suitability of the inpatient, presented the research and main objective, gave informed consent for authorisation to participate in the study and completed the self-completion questionnaire. Subsequently, we passed the scales.

Data collection was carried out using a Microsoft Access database created specifically for this study using a unique identification code for each participant in the study.

Statistical analysis was carried out using SPSS version 25.0 (IBM Corp., Armonk, NY, USA).

### 2.7. Statistical Analysis

The statistical analysis of the study involved the collection of data from the selected sample by means of a questionnaire and measurement scales, processing the data by applying exclusion criteria from the study when necessary.

We have resorted to descriptive analysis considering the maximum and minimum values obtained for each of the quantitative variables, as well as the presence of possible outliers in the box plots.

The determination of the outlying values and their consideration or not as part of the study has been done through the boxplot, considering the spread distance and classifying the outlying values into three types: adjacent values, close outlying values and far outlying values.

The far distant values were those that were determined to be removed from the study in the variables in order to avoid distortion by maintaining close distant values and adjacent values due to an ecological situation of patient participation in the study.

#### 2.7.1. Descriptive Statistics

We performed a descriptive analysis of the socio-demographic characteristics of our sample and of the scores we obtained using the study tools.

The study variables were analysed using the Shapiro-Wilk statistic to ascertain the normality of the sample, determining the path to follow: parametric (normal variables) or non-parametric (non-normal or ordinal variables).

In all cases we describe the variables with the corresponding statistics: the variables that have followed a normal distribution have been defined by mean and standard deviation statistics following the parametric route, and the variables that have followed a non-normal distribution have been defined by the median and the interquartile range as a measure of centralisation following the non-parametric route.

The categorical or qualitative variables were defined by frequencies and percentages.

#### 2.7.2. Analytical Statistics

The study variables were analysed using the Shapiro-Wilk statistic to determine whether they followed a normal distribution in each case and circumstance, thus determining the way forward: parametric (normal variables) or non-parametric (non-normal or ordinal variables).

The statistical consideration of the variables according to their distribution has been: mean and standard deviation (m and s) if we have normal variables, and median and Inter-quartile range (M and IQR) if we have non-normal variables.

The normality test of the aforementioned statistic oriented most of the calculations of the sample variables in a non-parametric way (*p* < 0.05) based on the fact that a comparison or correlation of two variables is carried out, except when the two variables are normal.

Some variables have been recoded as long as the number “N” of the variable under study was very small and had a coherent capacity to recode in another group of the same variable.

We used the Mann–Whitney U Test (for comparison of two means) or the Kruskal–Wallis Test (for comparison of three or more means) when the initial conditions were equal when *p* > 0.05.

The comparison of two means has been resolved in each case, considering, in the parametric way, with the Student’s *t*-test statistic (both in repeated means and in independent groups), in the non-parametric way with the Mann–Whitney U statistic (independent groups) or the Wilcoxon *t*-Test (repeated measures).

Comparisons of three or more means have been analysed: non-parametrically with the Kruskal–Wallis H statistic—ANOVA, non-parametric—in the situation of independent groups, and non-parametrically with Friedman’s Q test in the situation of repeated measures.

The results obtained have been expressed with the value of the statistic with the *p*-values and those data that are most interesting for the interpretation of the result.

The contrast statistics have been included for the most part in the analysis in order not to lose veracity in the statistical process and to be able to subtract information in the subsequent discussion independent of the normality test result (*p* < 0.005 or *p* > 0.005).

## 3. Results

The study had a final sample of 365 patients with a mean age of 61.62 years (±13.012).

Table 1 shows the socio-demographic characteristics of the patients with an equal distribution between men (48.5%) and women (51.5%), with 63.8% of the sample being married followed by 14.5% of the sample being single. The place of residence of the patients was 55.3% of the sample in an urban setting and 44.7% of the sample in a rural setting.

The type of cancer in the sample was 24.1% for the digestive system, followed by 23.3% for the respiratory system and lung, followed by 19.7% for the breast. The stage of the disease was stage IV in 30.4% of the sample, stage II in 28.2% of the sample, stage III in 24.75% of the sample, stage I in 14.0% and stage 0 in 2.2% of the sample.

The amount is reflected in the table of net annual household income during the last fiscal year when the patient was diagnosed and before the oncological diagnosis. The variation that the household has undergone since the cancer diagnosis is also reflected.

Descriptive results are shown for the extraordinary expenses incurred by the patients evaluated. Analysing the percentages, we can see that there has been an expense of up to €600 for 65.85% of the sample related to pharmacy and/or parapharmacy, an extraordinary expense of €600 and upwards for 21.4% of the sample related to orthopaedic material, an extraordinary expense of €600 and upwards for 7.7% of the sample on home help, and an extraordinary expense of €600 and upwards for 7.7% of the sample on home help, 7% of the sample for home help and/or patient accompaniment service and an extraordinary expense of more than €1500 on average for transfers to hospital for 11.2% of the sample, followed by an expense of €900 on average for 12.1% and up to €300 on average for 41.4% of the sample.

Table 2 shows the results obtained in the different measurement scales administered to the patients.

The Barthel Index score shows that 31.8% of the sample is independent, 35.9% has mild dependence, and 31.9% has moderate to total dependence.

The Lawton and Brody score shows that 24.7% of the sample is independent, 33.2% is mildly dependent, 20.8% is moderately dependent and 20.9% is severely to totally dependent.

The ECOG scale shows Ecog 1 for 34.2% of the sample, followed by Ecog 2 for 24.4% of the sample, Ecog 0 for 16.2% of the sample, and 24.7% corresponds to a situation between Ecog 3 and Ecog 5.

The Zarit scale shows that 65.8% of the main carers had no caregiving overload, followed by 20.3% of the sample with an intense overload and 13.4% with a light overload.

### Primary and Secondary Results

Table 3 shows the degree of statistical significance obtained in the contrast between the different economic variables and the grouping variables considered in the study.

Analysing the different variables under study, we can observe the following:–For the variable “Amount of net annual household income prior to cancer diagnosis” (*p* < 0.05) we observe from the average range how the amount of net annual household income prior to diagnosis is higher in the urban setting (192.03) than in the rural setting (169.70).–For the variable “Amount of net annual household income during the last fiscal year” (*p* < 0.05) we observe from the average range that the amount of net annual household income during the last fiscal year is higher when the type of cancer is digestive (192.85) or when it is another type of cancer (208.41). However, we can observe how the amount of net annual household income during the last fiscal year on average is much smaller when the patient has a haematological diagnosis (129.19).–For the variable “Extra expenditure in the last year on home help and/or patient accompaniment service” (*p* = 0.014) we observe from the average range how the extra expenditure in the last year on home help and/or patient accompaniment service is higher in the urban area (31.55) than in the rural area (21.61).–For the variable “Extra expense in transfers to hospital” (*p* = 0.021) we observe from the average range how the extra expense in transfers to hospital is higher when we have a diagnosis related to the Central Nervous System (151.15) or Haematology (145.77).

Table 4 shows the degree of statistical significance obtained in the contrast between the different measurement instruments and the grouping variables considered in the study.

Analysing the different variables under study, we can observe the following:–For the variable “Amount of net annual household income prior to cancer diagnosis” (*p* < 0.05) we observe how the Lawton and Brody scale shows us from the average range that there is a greater amount of income prior to cancer diagnosis when the patient is independent (201.44), followed by when the patient has a moderate score (188.63) and a light dependency score (177.43).–For the variable “Extraordinary expenditure in the last year on transfers to hospital” (*p* = 0.024), we observe that there is significance with the ECOG scale, which shows us from the ranges that there is greater expenditure when the patient has an Ecog 5 (201.25) with a large difference with respect to the other ranges of the same variable, being almost double with respect to Ecog 4 (106.91).–For the variable “Amount of net annual household income prior to diagnosis” (*p* = 0.029), we observe that there is significance with the ZARIT scale which shows us from the ranges that the amount of net household income prior to diagnosis is greater when the relative does not have caregiver overload (190.73), followed by when he/she has a slight overload (180.81).–For the variable “Amount of net annual household income during the last fiscal year” (*p* = 0.016) we observe that there is significance with the ZARIT scale which shows us from the ranges that the amount of net annual income during the last fiscal year in the household is greater if the main caregiver is in a situation without overload (191.21) followed by having a light overload (182.23) and having an intense overload (151.98).

## 4. Discussion

This research arose from the need to identify and analyse the additional cost and socioeconomic impact of cancer disease on patients and their families. Cancer generates an additional cost [11,12,13,14,15] that is not covered by the national health system, creating difficulties for the individual in the fight against cancer, not for medical but for economic reasons [13,14,15].

The evolutionary process of the disease not only generates additional cost with the possible physical state of disability or dependence of the patient, but, as shown by our results, additional costs related to the very fact of having cancer in terms of medicines, travel to receive treatment, consultations, home help and orthopaedic material, among others.

In recent years, although lifestyle changes are continually advocated to prevent the development of the disease, cancer rates worldwide continue to rise [23]. For many patients and families, coping with the disease involves socioeconomic inequalities, with low socioeconomic status being a factor directly associated with health disadvantage and increased mortality [24].

Access to adequate, high-quality, patient-centred cancer care is of vital importance to take into account all factors surrounding the sick person, including drug prices, acute hospitalisations, hospitalisations in the last months of life, among others [25]. The socioeconomic level of the family unit at the time of coping with the disease is directly related to better adherence to treatment and therefore a better quality of life [24,26].

The study by Wu et al. [26] reveals how even depressive symptoms in people with chronic diseases, in our case cancer patients, are associated with higher costs of medical care in outpatient visits and hospitalisation costs, but also with greater deterioration of their physical health, lower adherence to treatment and a decrease in quality of life with a corresponding economic burden on the patient due to physical limitations coexisting with depressive symptoms that exacerbate their morbidity and disability [27,28,29].

Chronic diseases have become an economic burden on health care worldwide [30,31,32]; patients present with mental and physical distress in their evolution [33] with chronic diseases such as cancer requiring long-term observation for decreased function and a greater episode of unpredictable acute care [34,35].

The economic circumstances of the patient and their family determine the financial stability to face certain costs; regardless of purchasing power, the families interviewed in our research have required a greater amount of economic resources during the process, as has occurred in other studies [36] in terms of medication, consultations, transport, special food, etc.

The results obtained in our study show that the main variables that entail additional expenses for the patient and their family are the cost of pharmacy and/or parapharmacy, the cost of orthopaedic material, the cost of home help and/or patient accompaniment service and the cost of transport to hospital, none of which is fully covered by Social Security.

These results coincide with studies such as that of Camacho and colleagues on amyotrophic lateral sclerosis (ALS) and neuromuscular diseases, in which they point out that their patients also require orthopaedic technical aids, assistance from a caregiver to carry out Activities of Daily Living, specific care needs, transport costs, etc., which are not funded [37]. As in our study, care needs represent an additional expense generating an economic impact on the patient and thus on the family itself.

Our results also coincide with the research of Villarejo and collaborators, who point out that Alzheimer’s disease (AD) and other dementias generate expenses derived from home care, institutionalisation, technical aspects, remodelling of homes, transport, informal caregiver expenses, etc., which are not financed by Social Security, with 88% of the total cost being borne by the patient’s family and only 12% being financed by public funds [38].

Regarding the amount of annual income of the patient and his family, the average values obtained from the study indicate that the amount of income in the last fiscal year is much lower when the patient is in a situation of diagnosis related to haematological cancer disease. This situation is due to the fact that the patient is admitted to hospital for long periods of time when receiving aplasia therapy and has to stay in hospital for three to five weeks, requiring specialised hospital care [39]; after discharge as well, the patient needs to stay in the vicinity of the hospital where he/she is being treated for any signs of fever or ailment in order to receive the corresponding antibiotic treatment.

Patients in this situation have to deal with the additional costs of having cancer, but also with situations of sick leave of the patient himself, together with the support and accompaniment of the main caregiver, generally a spouse or partner who also has to modify their work activity (sick leave, reduced working hours for care, months without employment or salary, leaving work, etc.), thus considerably affecting the economy of the family unit, which has to continue to meet the common expenses of their place of residence like any other family.

In this sense, the existing literature reinforces our findings and indicates that annual income is indeed reduced as a consequence of having an anatomopathological diagnosis of cancer. Studies such as that of Sharrocks and colleagues obtain similar results and affirm, when discussing clinical trials, the low representation of groups with a low socioeconomic level, which is an aspect that they relate to the financial capacity of the patient and the family who cannot cover additional expenses derived from a trial with travel to another place of residence different from their home, expenses in accommodation, diets, transport, etc. [40].

Regarding geographic dispersion, the results obtained in our research indicate that it generates additional expenses for the patient and the family for travelling to the hospital to receive treatment. The additional cost of transfers to hospital and/or patient accompaniment services, considering the average ranges, is higher when there is a diagnosis related to the central nervous system or haematological diagnosis compared to the rest of the oncological diagnoses.

These data coincide with the study by Villarejo et al. in relation to Alzheimer’s disease and other dementias, in which the patient’s place of residence is a factor to be considered; being registered in a rural or urban area is a major additional cost factor [38].

Our results are also similar to those reported by Wyman et al. who state that the families of cancer patients bear 45% of the total cost of the disease from their own resources in terms of pharmacy and/or parapharmacy, transport, food, accommodation, equipment, home works, formal and informal care, etc. [15].

In addition to the above findings, our results inform us that before the oncological diagnosis, there were more patients and family units with income in the range €120,001–18,000 (2.1% more), more patients with income in the range €180,001–24,000 (1.4%), and more patients with income of more than €30,000 (2.2% more). However, if there was a diagnosis of cancer, there were more patients and households with incomes below €6000 (1.1% more) and more patients and households with incomes in the range €6000–12,000 per year (2.2% more).

In short, according to the data obtained in our study, for 55% of patients with an annual net income of less than €15,000, cancer represents an extra expense of between 8.38–9.67% of their annual income.

Based on the results obtained, we can indicate that, indeed, the disease has an additional cost for the patient and the family that is not covered by the public sector simply because they have cancer, and, in addition, it represents a reduction in income. Other studies analysed show similar results [38], creating a new social reality of inequality and a social disadvantage in this case for the patients themselves.

The literature on this subject is still practically non-existent; however, we have found bibliography on the economic cost of cancer to the National Health System [41], the cost of cancer at hospital level with a public economic impact [42] or cancer related to the resulting incapacities in Social Security for the patient and loss of productivity [42].

It is important to point out that most research to date has focused primarily on the cancer disease and not on the person suffering from cancer in a comprehensive manner, taking into account the patient-family nucleus. For this reason, we consider that one of the great advantages of our work is to place the cancer patient at the epicentre of the intervention, considering all the variables that may affect the patient and the family in terms of having an additional cost.

The limitations of the study have been, firstly, the reliability of the sources: the subject of our research does not have sufficient theoretical and documentary support and, therefore, much of the information has been based on fieldwork through the techniques shown and tools retrieving information through informant subjects.

Another limitation is the study population and its contextual framework; the study participants are people diagnosed with cancer, and advocating empathetic behaviour at all times and absolute respect for the situation of the patient and their family can make it difficult to obtain such personal information.

In future studies, we intend to address different age groups affected by cancer in order to be able to discriminate whether, in more limited age groups, disability and dependence is incidental and can generate greater expenditure in addition to the additional expenditure involved in having cancer. We would also like to carry out a study differentiated by type of cancer in different age groups in order to be able to see the breakdown of expenses by type of cancer and which of them, in addition to being cancer, entail greater additional costs.

We are also considering the possibility of studying the cancer patient’s family unit; we start from the fact that the sample has a high rate of married patients. Studying the impact of cancer on the immediate family on socioeconomic levels (reduction of working hours, abandonment of work for care, unpaid leave, etc.) would be of interest to the oncology community. Our results lead us to think about the importance of focusing on the patient and not so much on the disease. In this way, different needs of patients, their families, and thus of our society, could be highlighted and reduced.

## 5. Conclusions

Cancer disease entails an additional cost for the patient and the family. In addition, the disability and dependence of patients does not represent an additional cost due to their levels of functionality, but it can have repercussions on the future cost of the evolution of the disease, in addition to the fact of having cancer.

## Figures and Tables

**Table 1 healthcare-10-02370-t001:** Socio-demographic variables of the study sample.

Socio-Demographic Variables	N	Results
Patient age	365	6162 ± 13.012
Patient gender	365	Men: 48.5%
		Women: 51.5%
Patient marital status	365	Single: 14.5%
		Married: 63.8%
		Separated and divorced: 11%
		Widowers: 10.7%
Patient place of residence	365	Urban: 55.3%
		Rural: 44.7%
Type of cancer	365	Digestive: 24.1%
		Lung: 23.3%
		Breast: 19.7%
		Prostate: 6.3%
		Central N.S.: 3.0%
		Haematological: 9.3%
		Other: 14.2%
Stage of the disease	363	Stage 0: 2.2%
		Stage I: 14.0%
		Stage II: 28.2%
		Stage III: 24.7%
		Stage IV: 30.4%
Number of treatments since cancer diagnosis	363	1.51 ± 0.653
Amount of annual net household income in the last tax year	363	Less than €6000: 4.9%
		From €6000 to €12,000: 22.2%
		From €12,001 to €18,000: 27.9%
		From €18,001 to €24,000: 20.0%
		From €24,001 to €30,000: 13.7%
		More than €30,000: 10.7%
Amount of net annual household income prior to cancer diagnosis	363	Less than €6000: 3.8%
		From €6000 to €12,000: 18.1%
		From €12,001 to €18,000: 30.1%
		From €18,001 to €24,000: 21.4%
		From €24,001 to €30,000: 13.2%
		More than €30,000: 12.9%
Change in annual net household income prior to cancer diagnosis	124	Up to 20%: 14.0%
		From 20% to 40%: 13.7%
		From 40% to 60%: 4.9%
		From 60% upwards: 1.4%
Extraordinary expenses in the last year in pharmacy and parapharmacy	274	Up to €600: 65.8%
		From €600 to €1200: 7.1%
		From €1201 upwards: 2.2%
Extraordinary expenditure on orthopaedic equipment in the last year	171	Up to €600: 25.5%
		From €600 to €1200: 16.2%
		From €1201 upwards: 5.2%
Extraordinary expenditure in the last year on home help and patient escort service	54	Up to €600: 7.1%
		From €600 to €1200: 3.6%
		From €1201 upwards: 4.1%
Average extraordinary expenditure on transfers to hospital in the last year	236	€300: 41.4%
		€900: 12.1%
		€1500: 4.4%
		€2400: 2.7%
		€4500: 4.1%

**Table 2 healthcare-10-02370-t002:** Scores on the IB, Lawton and Brody, ECOG and Zarit scales of the sample.

Measuring Scales	N	Results
Barthel score	363	Total: 13.2%
		Severe: 5.8%
		Moderate: 12.9%
		Slight: 35.9%
		Independent: 31.8%
Lawton and Brody score	363	Total: 8.8%
		Severe: 12.1%
		Moderate: 20.8%
		Slight: 33.2%
		Independent: 24.7%
Ecog scale	363	Ecog 0: 16.2%
		Ecog 1: 34.2%
		Ecog 2: 24.4%
		Ecog 3: 11.0%
		Ecog 4: 13.2%
		Ecog 5: 0.5%
Zarit scale	363	No overload: 65.8%
		Light overload: 13.4%
		Intense overload: 20.3%

**Table 3 healthcare-10-02370-t003:** Contrast statistics. Grouping variable: type of cancer and patient’s place of residence.

Variable	Type of Cancer	N	Value *p*	Average Range	Patient’s Place of Residence	N	Value *p*	Average Range
Amount of annual net household income prior to diagnosis	Digestive	88	*p* = 0.162	18,933	Urban	200	*p* = 0.039	19,203
	Lung	84						
	Breast	72		18,549				
	Prostate	23		18,092				
	Central N.S.	11		17,633	Rural	163		16,970
	Haematological	34		16,523				
	Other	51		13,725				
	Total	363		20,115	Total	363		
Amount of annual net household income in the last tax year	Digestive	88	*p* = 0.023	19,285	Urban	200	*p* = 0.72	19,075
	Lung	84						
	Breast	72		18,401				
	Prostate	23		17,585				
	Central N.S.	11		18,624	Rural	163		17,127
	Haematological	34		15,205				
	Other	51		12,919				
	Total	363		20,841	Total	363		
Change in revenue	Digestive	34	*p* = 0.054	5674	Urban	75	*p* = 0.899	6281
	Lung	25						
	Breast	21		6224				
	Prostate	7		6886				
	Central N.S.	5		4043	Rural	49		6203
	Haematological	16		5630				
	Other	16		8438				
	Total	124		5653	Total	124		
Extraordinary expenditure in the last year pharmacy and parapharmacy	Digestive	63	*p* = 0.292	13,977	Urban	156	*p* = 0.171	14,076
	Lung	66						
	Breast	53		14,370				
	Prostate	14		12,803				
	Central N.S.	7		12,050	Rural	118		13,319
	Haematological	31		12,050				
	Other	40		14,250				
	Total	274		14,130	Total	274		
Extraordinary expenditure in the last year on orthhopaedic equipment	Digestive	38	*p* = 0.129	7905	Urban	95	*p* = 0.865	8548
	Lung	36						
	Breast	47		8728				
	Prostate	9		8428				
	Central N.S.	6		7233	Rural	76		8664
	Haematological	12		13,633				
	Other	23		8825				
	Total	171		9004	Total	171		
Extraordinary expenditure in the last year on home help and patient accompaniment service	Digestive	19	*p* = 0.879	2700	Urban	32	*p* = 0.014	3155
	Lung	8						
	Breast	7		3094				
	Prostate	3		2307				
	Central N.S.	2		3583	Rural	22		2161
	Haematological	3		2325				
	Other	12		2467				
	Total	54		2792	Total	54		
Extraordinary expenditure on transfers to hospital	Digestive	58	*p* = 0.021	10,834	Urban	97	*p* = 0.845	11,939
	Lung	56						
	Breast	43		11,452				
	Prostate	15		11,569				
	Central N.S.	10		9187	Rural	139		11,788
	Haematological	26		15,115				
	Other	28		14,577				
	Total	236		12,911	Total	236		

**Table 4 healthcare-10-02370-t004:** Contrast statistics. Grouping variable: measurement scales.

Variable	Barthel	N	Value *p*	Lawton Brody	N	Value *p*	ECOG	N	Value *p*	ZARIT	N	Value *p*
Amount of annual net household income prior to diagnosis	Total	48	*p* = 0.186	Total	32	*p* = 0.046	Ecog 0	59	*p* = 0.194	No overload	240	*p* = 0.029
	Severe	21		Severe	44		Ecog 1	125				
	Moderate	47		Moderate	76		Ecog 3	89		Light overload	49	
	Slight	131		Light	121		Ecog 4	48				
	Independent	116		Independent	90		Ecog 5	2		Intense overload	74	
	Total	363		Total	363		Total	363		Total	363	
Amount of annua net household income in the last tax year	Total	48	*p* = 0.475	Total	32	*p* = 0.081	Ecog 0	59	*p* = 0.424	No overload	240	*p* = 0.016
	Severe	21		Severe	44		Ecog 1	125				
	Moderate	47		Moderate	76		Ecog 3	89		Light overload	49	
	Slight	131		Light	121		Ecog 4	48				
	Independent	116		Independent	90		Ecog 5	2		Intense overload	74	
	Total	363		Total	363		Total	363		Total	363	
Change in revenue	Total	7	*p* = 0.540	Total	8	*p* = 0.273	Ecog 0	14	*p* = 0.152	No overload	74	*p* = 0.614
							Ecog 1	50				
	Severe	10		Severe	11		Ecog 2	36		Light overload	17	
	Moderate	17		Moderate	32		Ecog 3	13				
	Slight	51		Light	48		Ecog 4	10		Intense overload	33	
	Independent	39		Independent	25		Ecog 5	1				
	Total	124		Total	124		Total	124		Total	124	
Extraordinary expenditure in the last year pharmacy and parapharmacy	Total	36	*p* = 0.663	Total	27	*p* = 0.957	Ecog 0	35	*p* = 0.328	No overload	179	*p* = 0.095
							Ecog 1	99				
	Severe	15		Severe	33		Ecog 2	67		Light overload	35	
	Moderate	43		Moderate	60		Ecog 3	36				
	Slight	102		Light	94		Ecog 4	35		Intense overload	60	
	Independent	78		Independent	60		Ecog 5	2				
	Total	274		Total	274		Total	274		Total	274	
Extraordinary expenditure in the last year on orthhopaedic equipment	Total	39	*p* = 0.451	Total	26	*p* = 0.142	Ecog 0	13	*p* = 0.693	No overload	92	*p* = 0.302
							Ecog 1	48				
	Severe	11		Severe	29		Ecog 2	42		Light overload	32	
	Moderate	35		Moderate	43		Ecog 3	27				
	Slight	52		Light	43		Ecog 4	39		Intense overload	47	
	Independent	34		Independent	31		Ecog 5	2				
	Total	171		Total	171		Total	171		Total	171	
Extraordinary expenditure in the last year on home help and patient accompaniment service	Total	11	*p* = 0.766	Total	7	*p* = 0.955	Ecog 1	12	*p* = 0.090	No overload	30	*p* = 0.621
	Severe	4		Severe	8							
	Moderate	14		Moderate	23		Ecog 2	21		Light overload	7	
	Slight	20		Light	15		Ecog 3	10				
	Independent	5		Independent	1		Ecog 4	11		Intense overload	17	
	Total	54		Total	54		Total	54		Total	54	
Extraordinary expenditure on transfers to hospital	Total	37	*p* = 0.885	Total	26	*p* = 0.850	Ecog 0	32	*p* = 0.024	No overload	146	*p* = 0.806
							Ecog 1	72				
	Severe	16		Severe	34		Ecog 2	65		Light overload	33	
	Moderate	31		Moderate	48		Ecog 3	26				
	Slight	92		Light	81		Ecog 4	39		Intense overload	57	
	Independent	60		Independent	47		Ecog 5	2				
	Total	236		Total	236		Total	236		Total	236	

## Data Availability

The data presented in this study are available on reasonable request from the corresponding author. The data are not publicly available due to the applicable data protection law.

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
