# Peer review of "Study on the Socio-Economic Impact of Cancer Disease on Cancer Patients and Their Relatives"

_healthcare, 2022, doi:10.3390/healthcare10122370_

Round 1

Reviewer 1 Report

The article presents new, original and relevant research study on the additional/hidden costs of cancer, related to various inequalities in the level of patient access to medical services, due to a variety of socio-economic factors. However, I have a few more formal remarks that may significantly improve the clarity of the description of the method and results of the study conducted. 1) Why did certain variables have a non-normal distribution? 2) The 'Discussion' section already contains some conclusions and closes with the wording 'To conclude...', meanwhile it is followed by the 'Conclusions' section, which is precisely where conclusions should be found - and in a structured, more complete form than the current 'Conclusions'. The 'Discussion' section, on the other hand, is usually intended to describe the various limitations and shortcomings of the study, for counter-arguments and polemics against the methods used or the results obtained. In short, there is a mix of content in the 'Conclusions' and 'Discussion' sections. 3) There are many tabular summaries in the article; descriptive explanations precede the tables, while it would have been more skillful to place them below the tables. 4) There are linguistic errors, e.g. 'anual' or 'annua' in the table on p. 10, 'urbano' instead of 'urban' on p. 11, etc.; the names of items in the tables do not exactly correspond with the names of the same items in the descriptions of variables and research results – please adjust them. I recommend the manuscript for publication, however, only after all these minor revisions have been taken into account by the authors. Congratulations. 

Author Response

POINT-BY-POINT RESPONSE TO REVIEWERS

REVIEWER 1

Response: Thank you very much for your comment.

Reviewers: 1) Why did certain variables have a non-normal distribution?

Response: We understand that the statistical non-normality of some variables is due to the fact that there are some extreme values, conditioned, for example, because in some people the economic impact is very high due to their individual characteristics (for example, living in a different city from where the oncological treatment is carried out).

2) The 'Discussion' section already contains some conclusions and closes with the wording 'To conclude...', meanwhile it is followed by the 'Conclusions' section, which is precisely where conclusions should be found - and in a structured, more complete form than the current 'Conclusions'. The 'Discussion' section, on the other hand, is usually intended to describe the various limitations and shortcomings of the study, for counter-arguments and polemics against the methods used or the results obtained. In short, there is a mix of content in the 'Conclusions' and 'Discussion' sections.

Response: Thank you very much for your comment. We fully agree with your input, we proceed to modify the structure so that each section is limited to its content.

3) There are many tabular summaries in the article; descriptive explanations precede the tables, while it would have been more skillful to place them below the tables.

Response: Thank you very much for your comment. We have modified this in the manuscript.

4) There are linguistic errors, e.g. 'anual' or 'annua' in the table on p. 10, 'urbano' instead of 'urban' on p. 11, etc.; the names of items in the tables do not exactly correspond with the names of the same items in the descriptions of variables and research results – please adjust them.

Response: Thank you very much for your comment. We have modified this in the manuscript.

I recommend the manuscript for publication, however, only after all these minor revisions have been taken into account by the authors. Congratulations. 

Response: Thank you very much for your comment.

Reviewer 2 Report

I admire the authors for undertaking this impressive work and presenting it in a well-scientific manner. However, here are my comments which should be addressed before further consideration: 

1. Introduction: There are so many smaller paragraphs that should be merged. Please make 3 paragraphs with separate objectives of the study. 

2. Methods: Please add the Institutional Review Board Statement information here. 

Author Response

POINT-BY-POINT RESPONSE TO REVIEWERS

REVIEWER 2

Reviewers: I admire the authors for undertaking this impressive work and presenting it in a well-scientific manner. However, here are my comments which should be addressed before further consideration: 

Response: Thank you very much for your comment.

Reviewers: 1. Introduction: There are so many smaller paragraphs that should be merged. Please make 3 paragraphs with separate objectives of the study. 

Response: Thank you very much for your comment. We have made the modifications you have suggested, which will improve the comprehensibility of the manuscript.

Reviewers: 2. Methods: Please add the Institutional Review Board Statement information here. 

Response: Thank you very much for your comment. We have added the information in the methods section.

Reviewer 3 Report

This is a very interesting study which gives important information. This research area is rare, particularly in Europe. Methods, analyses, results are extensively and carefully reported.

I have only two remarks: 1- for quantitative variables it is better to give median (instead of mean) and extremes.; 2- the abstract is not informative: background “The disease entails additional costs for cancer patients and their families” and conclusion “Cancer disease entails an additional 23 cost for the patient and family”. I think that the abstract needs to include several concrete data.

Author Response

POINT-BY-POINT RESPONSE TO REVIEWERS

REVIEWER 3

Reviewers: This is a very interesting study which gives important information. This research area is rare, particularly in Europe. Methods, analyses, results are extensively and carefully reported.

Response: Thank you very much for your comment.

Reviewers: 1- for quantitative variables it is better to give median (instead of mean) and extremes.

Response: Thank you very much for your comment. We have made the modification you have suggested. We have modified the mean scores of quantitative values to median scores. We note that most scores are average ranges. However, we have carried out a detailed analysis of what is at stake.

Reviewers: 2- the abstract is not informative: background “The disease entails additional costs for cancer patients and their families” and conclusion “Cancer disease entails an additional 23 cost for the patient and family”. I think that the abstract needs to include several concrete data.

Response: Thank you very much for your comment. We have added more specific information in the "conclusions" section of the abstract.
